# Lysosomal Network Defects in Early-Onset Parkinson’s Disease Patients Carrying Rare Variants in Lysosomal Hydrolytic Enzyme Genes

**DOI:** 10.3390/ijms26199454

**Published:** 2025-09-27

**Authors:** Alba Pascual, Thaleia Moulka, Oriol de Fàbregues, Roberta Repossi, Pedro J. García-Ruiz, Saida Ortolano, Marisel De Lucca, Lydia Vela-Desojo, Marta Alves-Villar, Marcos Frías, Cici Feliz-Feliz, Mònica Roldán, Jonathan Olival, Guerau Fernàndez, Francesc Palau, Jordi Pijuan, Janet Hoenicka

**Affiliations:** 1Laboratory of Neurogenetics and Molecular Medicine, Center for Genomic Sciences in Medicine, Institut de Recerca Sant Joan de Déu, 08950 Barcelona, Spainfrancesc.palau@sjd.es (F.P.); jordi.pijuan@sjd.es (J.P.); 2Neurodegenerative Diseases Research Group, Vall d’Hebron Research Institute, Center for Networked Biomedical Research on Neurodegenerative Diseases (CIBERNED), 08035 Barcelona, Spain; 3Movement Disorders Unit, Neurology Department, Vall d’Hebron University Hospital, 08035 Barcelona, Spain; 4Unit of Movement Disorders, Department of Neurology, Fundación Jimenez Díaz, 28040 Madrid, Spain; 5Rare Diseases & Pediatric Medicine Research Group, Galicia Sur Health Research Institute (IIS Galicia Sur), Servicio Gallego de Salud-Universidad de Vigo (SERGAS-UVIGO), 36213 Vigo, Spain; saida.ortolano@iisgaliciasur.es (S.O.); marta.alves@iisgaliciasur.es (M.A.-V.); 6Department of Biological Sciences, Faculty of Health Sciences, Universidad Técnica de Manabí, Portoviejo 130105, Ecuador; marisel.delucca@utm.edu.ec; 7Movement Disorders Unit, Department of Neurology, Hospital Universitario Fundación Alcorcón, 28922 Madrid, Spain; 8Confocal Microscopy and Cellular Imaging Unit, Hospital Sant Joan de Déu, 08950 Barcelona, Spain; 9Department of Genetics, Hospital Sant Joan de Déu, 08950 Barcelona, Spain; 10Centro de Investigación Biomédica en Red de Enfermedades Raras (CIBERER), Instituto de Salud Carlos III (ISCIII), 28029 Madrid, Spain; 11Hospital Sant Joan de Déu, 08950 Barcelona, Spain; 12Division of Pediatrics, Faculty of Medicine and Health Sciences, University of Barcelona, 08036 Barcelona, Spain

**Keywords:** early-onset Parkinson’s disease (EOPD), α-galactosidase A, *GLA*, β-galactosidase, *GLB1*, lysosomal dysfunction

## Abstract

Despite significant advances in understanding the genetics of Parkinson’s disease (PD) and Parkinsonism, the diagnostic yield remains low. Pathogenic variants of *GBA1*, which encodes the lysosomal enzyme β-glucocerebrosidase and causes recessive Gaucher dis-ease, are recognized as the most important genetic risk factor for PD in heterozygous carriers. This study focuses on the functional genomics of rare genetic variations in other lysosomal hydrolytic enzymes genes in patient-derived fibroblasts. We examined 49 early-onset PD patients using whole exome sequencing and in silico panel analysis based on a curated PD gene list. Two patients were found to carry the p.Asp313Tyr variant in the X-linked *GLA* gene (encoding GALA, typically associated with Fabry disease), and one patient carried the p.Arg419Gln variant in *GLB1* (encoding β-Gal, linked to the recessive GM1 gangliosidosis and mucopolysaccharidosis type IVB). The in silico study of both variants supports a potentially damaging impact on the encoded protein function and structural destabilization. Additional candidate variants were found related to lysosomes, Golgi apparatus and neurodegeneration, suggesting a multifactorial contribution to the disease. However, none of these variants met diagnostic standards. Functional assays showed a significant decrease in GALA expression and partial retention of the enzyme in the trans-Golgi network in fibroblasts with *GLA*:p.Asp313Tyr, while altered Golgi morphology was observed in fibroblasts with *GLB1*:p.Arg419Gln. Moreover, all patients exhibited abnormalities in lysosomal morphology, altered lysosomal pH, and impaired autophagic flux. Our findings suggest that rare, heterozygous variants in lysosomal-related genes, even when individually insufficient for monogenic disease, can converge to impair lysosomal homeostasis and autophagic flux in EOPD. The underlying genetic and cellular heterogeneity among patients emphasizes the importance of combining genetic and functional approaches to better understand the mechanisms behind the EOPD, which could enhance both diagnosis and future treatments.

## 1. Introduction

Parkinson’s disease (PD) is an age-related movement and neurodegenerative disorder characterized by a progressive loss of dopaminergic neurons in the substantia nigra pars compacta (SNpc) and the accumulation of Lewy bodies [1]. This degeneration results in a gradual dopaminergic deficiency, disrupting brain neurotransmitter systems. Some patients develop Parkinsonism before age 50, which is known as early-onset Parkinson’s disease (EOPD) [1]. The etiological diagnosis of EOPD patients increasingly identifies less common genetic disorders that cause parkinsonism in younger individuals. However, the success of genetic diagnosis in both EOPD and PD remains limited, and a significant portion of heritability for Parkinsonisms remains unexplained [2], highlighting the need to further explore both known and novel genetic contributors.

The genetic architecture of Parkinsonisms ranges from rare, high-penetrance variants—often responsible for familial, monogenic forms—to common variants with modest effects that contribute to sporadic cases and may influence disease expression [3]. Growing evidence strongly links the dysfunction of the autophagy and endolysosomal pathways with the development of PD [4,5]. Lysosomes are essential for maintaining neuronal proteostasis by degrading cellular debris via autophagy and endocytosis. When this process is defective, misfolded proteins like α-synuclein accumulate, leading to neuronal toxicity and neurodegeneration [5]. Additionally, a higher load of genetic variants in lysosomal storage disorder (LSD)-related genes has been observed in PD patients [6]. Notably, pathogenic variants in the *GBA1* gene (MIM*606463), which encodes the lysosomal enzyme β-glucocerebrosidase (GCase) and causes recessive Gaucher disease, are recognized as the most significant genetic risk factor for PD in heterozygous carriers [7,8]. *GBA1* variants not only increase PD risk but are also linked to a more severe disease progression [9]. In addition to *GBA1*, several other lysosomal genes—including *ATP13A2*, *LAMP1*, *TMEM175*, *VPS13C*, *SMPD1*, *SCARB2*, and *GALC*—have been found to carry a higher load of rare variants in PD patients [10,11]. Moreover, multiple lines of evidence connect LSDs and PD: (i) neurodegeneration is a common feature of many LSDs; (ii) Parkinsonism symptoms such as tremor, bradykinesia, and rigidity have been observed in patients with certain LSDs; and (iii) α-synuclein aggregation has been detected in both human and animal models of several LSDs [12,13].

Other LSD genes that have recently gained attention in PD studies are *GLA* (MIM*300644) and *GLB1* (MIM*611458), which encode α-galactosidase (GALA) and β-galactosidase (β-GAL), respectively. These enzymes are part of the lysosomal sphingolipid-degrading pathway that also involves GCase (Figure 1). Pathogenic variants in *GLA* cause Fabry disease (FD) (MIM#301500), an X-linked LSD characterized by deficient GALA activity and subsequent accumulation of glycosphingolipids such as globotriaosylceramide (Gb3) and globotriaosylsphingosine (lyso-Gb3). Although the evidence remains limited, Parkinsonism has been reported in some individuals with FD, suggesting a possible clinical and genetic overlap with PD [5,14]. Furthermore, PD patients with α-GAL deficiency have been documented [14,15,16,17,18,19,20], and *GLA* variants are overrepresented among PD patients [5,14].

Similarly, pathogenic biallelic variants in *GLB1* cause Mucopolysaccharidosis type IVB (MIM#253010) and GM1-gangliosidosis types I–III (MIM#230500, 230600, 230650), both LSD characterized by the lysosomal accumulation of GM1 gangliosides. These conditions result from deficient β-GAL activity and cause various neurological symptoms. Notably, Parkinsonian features have been described in patients with compound heterozygous *GLB1* variants, further supporting the involvement of lysosomal dysfunction in PD pathogenesis [21,22].

This study aims to investigate further how rare genetic variations in lysosomal genes encoding hydrolytic enzymes contribute to EOPD susceptibility. Using whole-exome sequencing (WES), we identified patients with variants in the *GLA* and *GLB1* genes. Functional analyses revealed common alterations in lysosomal biology and autophagy, highlighting the genetic and cellular diversity of EOPD.

## 2. Results

### 2.1. Clinical and Genetic Findings

We conducted WES for genomic analysis in 49 patients with EOPD, all of whom had previously tested negative for PD-associated genes using a custom panel [23]. WES was performed using different strategies, including singleton and trio (family-based) sequencing. A list of 3589 genes was compiled, which included genes for the following conditions: PD [1,24]; atypical Parkinsonism [25], other movement disorders [26,27,28], and treatable inherited rare movement disorders [29]. The PD gene list was expanded with genes identified in panels for Parkinsonism or other movement disorders from PanelApp [30], genes involved in copper metabolism [31], genes related to tau pathology disorders [32], and genes involved in mitochondrial and lysosomal functions from the Human Gene Mutation Database (HGMD) [33]. This gene list was further augmented using the genepanel.iobio.io generation list tool, which creates gene panels based on suspected conditions and phenotypes [34]. Finally, we included genes associated with lysosomal function and structure, phagosomes, the Golgi apparatus, and autophagy. All genes were reviewed according to the standards of the HUGO Gene Nomenclature Committee (HGNC) [35]. Patients’ WES data were filtered, and genes from the PD gene list were examined. Knowledge-based analysis (KDA) was then used for variant prioritization (Appendix A), resulting in a personalized score. This process identified three unrelated patients with rare variants in genes encoding lysosomal hydrolases (Table 1). Three variants per case are shown, prioritized by their lysosomal signaling pathway, clinical significance, and overall KDA score. Additionally, full candidate variant information can be found in Appendix A. None of the WES findings were diagnostic for any of the cases.

The first patient (PD-302) carries the benign (B) p.Asp313Tyr missense variant in *GLA* inherited from his mother, and two heterozygous variants of uncertain significance (VUS) in the PD genes *FBXO7*/PARK15 (autosomal recessive inheritance) and *LRRK2*/PARK8. The patient was diagnosed with PD at age 34. He presented with progressive asymmetric Parkinsonism, responsive to levodopa. Still, he developed severe motor fluctuations and dystonic “off” periods, requiring deep-brain stimulation (DBS) of the bilateral subthalamic nuclei at age 43. DaTSCAN imaging showed nigrostriatal dysfunction, and brain MRI revealed hyperintense demyelinating lesions (Appendix A). There was no reported family history of PD.

The second patient (PD-227) has the same *GLA* variant as the previous case, inherited from his mother. He is also a heterozygous carrier of a VUS variant in *BORCS8* linked to the recessive condition infantile-onset neurodegeneration with optic atrophy and brain abnormalities (NDOABA, MIM#620987), as well as a likely pathogenic (LP) variant in the *ABL2* gene without a MIM phenotype. Interestingly, *BORCS8* encodes a protein that regulates lysosome positioning [36]. PD-227 was diagnosed at age 40. He first presented with restless legs syndrome and generalized akathisia, followed by mild bilateral bradykinesia and gait freezing. Bouts of neuropathic pain in the lower limbs were treated with gabapentin. DaTSCAN revealed a moderate presynaptic dopaminergic deficit at the level of both the putamen and caudate nuclei. He has been treated with DOPA and apomorphine with moderate improvement.

The third male patient (PD-216) was heterozygous for three lysosomal-related genes. The LP variant p.Arg419Gln in the *GLB1* gene was inherited from his father; the VUS variant in the *VPS33B* gene, involved in Golgi-to-lysosome trafficking and associated with several recessive diseases (MIM#208085, MIM#620010, MIM#620009), inherited from his mother; and another VUS variant in a gene without a MIM phenotype. PD-216 was diagnosed with PD at age 37. His initial symptoms included left-sided bradykinesia and rigidity. From the beginning, he was managed with a combination therapy, including low-dose levodopa, which provided a moderate clinical response. He later underwent high-intensity focused ultrasound targeting the subthalamic nucleus, with only mild improvement. Ten years after diagnosis, he began experiencing motor fluctuations, mainly characterized by lower limb dystonia, dysarthria, and rigidity during “off” periods. Subcutaneous apomorphine infusion was initiated but had to be discontinued due to intolerable somnolence. Currently, he takes five daily doses of levodopa along with amantadine, rasagiline, and rotigotine. Notably, he has never developed dyskinesias. He experienced hypersexuality while on 10 mg of rotigotine, which resolved after reducing the dose to 8 mg. He also developed hobbyism (compulsive engagement in activities) for a few months. There is no family history of PD.

Since two patients carried the *GLA*:p.Asp313Tyr variant, we reviewed data on GALA activity and *GLA* variants in PD clinical series. We found 9 cohort studies on PD patients. Some articles indicated GALA deficiencies in PD patients, and an overrepresentation of *GLA*:p.Asp313Tyr (Appendix A). Considering this evidence from several independent studies pointing to the involvement of the *GLA*:p.Asp313Tyr in PD vulnerability and the predicted pathogenicity of *GLB1*:p.Arg419Gln, we selected these candidate variants for detailed in silico analysis. According to the American College of Medical Genetics and Genomics (ACMG) guidelines for variant pathogenicity prediction [37], *GLA*:p.Asp313Tyr is currently classified as B (PP2, BS1, BS2, BP6; last accessed August 2025). However, it was previously considered pathogenic (first accessed January 2023). Currently, *GLB1*:p.Arg419Gln is classified as LP (PM1, PP2, PM2, PM5, PP5; last accessed August 2025). Other indicators of pathogenicity produced inconsistent results (Appendix A). Furthermore, the *GLA*:p.Asp313Tyr variant shows discrepancies in reported allele frequencies across population databases. *GLB1*:p.Arg419Gln exhibits extremely low allele frequency in control populations such as gnomAD and CSVS (Appendix A). In silico structural analyses suggest that both the *GLA*:p.Asp313Tyr and *GLB1*:p.Arg419Gln amino acids substitutions would affect their respective proteins. They are highly conserved across phylogenetically distant species, indicating potential functional importance. The structural analysis based on crystallographic data (PDB:1R46 for GALA and PDB:3THC for β-GAL) revealed that both variants cause local changes in the secondary structure of the proteins (Figure 2A,E). Additionally, B-factor analyses showed decreased atomic mobility at the variation sites in both *GLA*:p.Asp313Tyr and *GLB1*:p.Arg419Gln, suggesting increased local rigidity compared to wild-type proteins (Figure 2B,F). Electrostatic surface potential maps also showed significant differences between wild-type and mutant forms of both proteins (Figure 2C,G). Finally, stability predictors (DynaMut2, DUET, and FoldX) consistently indicated a destabilizing effect for both variants. GALA^p.Asp313Tyr^ was associated with a notable decrease in folding free energy (*ΔΔG*), indicating substantial structural destabilization. Similarly, β-GAL^p.Arg419Gln^ showed negative *ΔΔG* values, supporting a predicted loss of protein stability (Figure 2D,H). Overall, these in silico results support a potential damaging impact of both *GLA*:p.Asp313Tyr and *GLB1*:p.Arg419Gln on the encoded protein function and structural destabilization.

### 2.2. Patient Carrying the GLA:p.Asp313Tyr Variant Shows Partial Retention of GALA in the Trans-Golgi Network

To evaluate the functional impact of *GLA*:p.Asp313Tyr, we initially measured GALA and GCase enzyme activities in leukocytes from patient PD-302. The patient’s GALA activity was within the normal range (0.71 nmol/min/mg protein; normal range: 0.25–0.94). However, we found that GCase activity was slightly elevated in the patient (0.44 nmol/min/mg protein; normal range: 0.14–0.36). We then examined the levels of globotriaosylceramide (Gb3), a marker in FD patients where it accumulates due to the lack of GALA activity [38,39]. Measurement of Gb3 levels in PD-302 fibroblasts showed no significant differences compared to controls (Appendix A). These results suggest that the enzymatic function of GALA might be partially preserved or compensated in the patient’s leukocytes by increased GCase activity.

To measure GALA expression in the patient’s fibroblasts, we conducted Western blot analysis of total protein extracts. We observed that GALA levels remained consistent in the patient (Figure 3A). Since GALA is mainly localized in lysosomes and also passes through the Golgi apparatus for post-translational maturation and transport to lysosomes [40], we examined its colocalization with the lysosomal membrane marker LAMP1 and Golgi markers. No significant differences in colocalization with LAMP1 were found (Figure 3B), but there was a notable increase in colocalization with the trans-Golgi marker TGN46 (CT: 0.36 ± 0.12 vs. PD-302: 0.45 ± 0.15; *p* < 0.013) (Figure 3C). Partial retention of GALA in the trans-Golgi network could disrupt its proper transport to lysosomes, suggesting a possible alteration in GALA biology in the PD-302 patient carrying the *GLA*:p.Asp313Tyr variant.

### 2.3. The Patient Carrying GLB1:p.Arg419Gln Shows Trans-Golgi Network Defects

Next, we examined the effect of *GLB1*:p.Arg419Gln on β-GAL expression in the PD-216 patient’s fibroblasts. β-GAL immunostaining revealed no significant differences in expression levels (Figure 4A). Because β-GAL undergoes post-translational processing and trafficking through the Golgi apparatus [41], we assessed its colocalization with TGN46. Manders’ colocalization coefficient values indicated no significant differences between PD-216 and control fibroblasts (Figure 4B), although the Golgi apparatus morphology was disrupted in the patient. We measured several morphological parameters in PD-216 and control fibroblasts, including: (i) the percentage of cells with fragmented versus unfragmented Golgi, (ii) the distance between the nucleus and the Golgi apparatus, and (iii) the number of Golgi fragments per cell. PD-216 patient showed a significantly higher percentage of cells with fragmented or beehive Golgi morphology (82.52%) compared to controls (14.67%) (*p* < 0.001). The distance from the Golgi to the nucleus, which reflects Golgi expansion and disorganization, was also significantly greater in PD-216 fibroblasts (CT: 5.98 ± 2.9 µm vs. PD-216: 12.01 ± 5.0 µm; *p* = 0.0001). Finally, the number of Golgi fragments per cell, indicating Golgi disaggregation, was significantly higher in the patient than in the controls (CT: 4.12 ± 2.83 vs. PD-216: 12.25 ± 11.8; *p* = 0.03) (Figure 4C). These findings suggest that *GLB1*:p.Arg419Gln may induce structural changes in the Golgi apparatus, particularly in the trans-Golgi network, which could impact cellular trafficking and organelle stability.

### 2.4. Variants in Genes Encoding Lysosomal Hydrolases Are Linked to Changes in Lysosomal Morphology and pH Levels

Next, we performed morphological studies by super-resolution confocal microscopy in fibroblast samples immunostained with α-LAMP1. We evaluate lysosomal size and lysosomal network porosity in patients PD-302 and PD-216. We also evaluated a third patient with EOPD (PD-212) who carries a genetic variant in *GBA1*:p.Leu483Pro, and a fourth patient with EOPD but no identified variants in lysosomal genes (PD-088; *PRKN*:p.Trp445*/DEL exon 3-6). We hypothesized that lysosomal defects are a common cellular feature in patients with genetic variants in lysosomal enzyme genes.

The comparison of lysosome size among samples revealed significantly larger sizes in patients with lysosomal gene variants (PD-302, PD-216, and PD-212), but not in PD-088, who is a carrier of *PRKN* biallelic pathogenic variants, compared with controls (CT: 482.9 ± 111.2 vs. PD-302: 645.8 ± 116.7 (*p* < 0.0001); PD-216: 749.1 ± 175.2 (*p* < 0.0001); PD-212: 588.2 ± 134.7 (*p* = 0.0002); PD-088: 443.0 ± 111.7) (Figure 5B). Regarding the lysosomal network porosity, which measures the distribution and density of lysosomes within the cell, we also see significant differences among samples. Patients with lysosomal gene variants exhibited significantly higher porosity, while the PD-088 patient’s lysosomal network was similar to that of the controls (CT: 12.83 ± 14.07 vs. PD-302: 48.57 ± 26.87 (*p* < 0.0001); PD-216: 64.59 ± 27.87 (*p* < 0.0001); PD-212: 55.50 ± 27.26 (*p* < 0.0001); PD-088: 18.29 ± 18.04) (Figure 5C). Higher lysosomal network porosity indicates decreased clustering, lower density, and abnormal morphology. To determine if these structural changes affected lysosomal lumen function, we measured lysosomal pH in vivo using LysoSensor Green, which emits its strongest fluorescence at the acidic pH range necessary for proper lysosomal enzyme activity (pH 4.5–5.0). We observed different pH levels in patients with lysosomal gene variants compared to controls and PD-088 (CT: 0.99 ± 0.08 vs. PD-302: 0.91 ± 0.13 (*p* < 0.001); PD-216: 0.98 ± 0.11 (*p* = 0.03); PD-212: 1.08 ± 0.11 (*p* < 0.001); PD-088: 1.03 ± 0.11). Specifically, PD-302 and PD-216 showed lower LysoSensor signals, indicating increased lysosomal pH (less acidic), which is linked to impaired lysosomal and autophagic functions. Conversely, PD-212 (*GBA1*:p.(Leu483Pro)) had a more acidic lysosomal pH, consistent with prior findings that certain *GBA1* variants can cause excessive acidification [42] (Figure 5D,E). Overall, these results suggest that genetic variants in lysosomal hydrolase genes lead to changes in lysosomal morphology and pH, potentially disrupting the activity of hydrolases and autophagy. The larger size, higher porosity, and altered acidity of lysosomes demonstrate a common pattern of lysosomal dysfunction in patients with PD who carry these variants.

### 2.5. Impaired Autophagic Flux in Patients with Variants in Lysosomal Genes

Finally, we aimed to assess autophagic flux in the patients’ fibroblasts. In PD, the autophagy-lysosomal pathway, a cellular process responsible for degrading proteins and organelles, is impaired, leading to the accumulation of toxic α-synuclein aggregates and neuronal cell death. This dysfunction, affecting both autophagy and lysosomal activity, contributes to the progression of PD [43]. We then examined autophagy in fibroblasts from patients PD-302, PD-216, PD-212, and PD-088, and compared them with healthy controls. Autophagy was stimulated by nutrient deprivation using Earle’s balanced salt solution (EBSS), and autophagic activity was measured by Western blot analysis of the following markers: Sequestosome-1/p-62 (an autophagic degradation marker), LC3-II/LC3-I ratio (an autophagosome formation marker), and LAMP1 (a lysosomal membrane marker).

Under normal conditions, a significant decrease in LAMP1 levels was observed in fibroblasts from two patients, PD-302 and PD-216, compared to the controls (CT: 0.98 ± 0.08 vs. PD-302: 0.57 ± 0.08 (*p* = 0.002); PD-216: 0.51 ± 0.20 (*p* = 0.0008)). Similarly, p-62 protein levels were lower in these two patients (CT: 0.98 ± 0.01 vs. PD-302: 0.51 ± 0.13 (*p* = 0.004); PD-216: 0.63 ± 0.05 (*p* = 0.02)), and the LC3-II/LC3-I ratio was slightly reduced in PD-302 (CT: 0.96 ± 0.04 vs. PD-302: 0.85 ± 0.19 (*p* = 0.007)), indicating subtle changes in autophagosome dynamics under baseline conditions (Figure 6A). After EBSS treatment, control fibroblasts responded as expected, with a notable decrease in p-62 levels indicating active autophagic flux. However, no significant changes in p-62, LC3-II/LC3-I, or LAMP1 levels were observed in any of the patients’ fibroblasts following starvation (Figure 6B). These findings suggest a failure to induce autophagy in response to nutrient deprivation, pointing to a dysfunction in the autophagic machinery.

To further investigate autophagic flux, we measured p62 aggregates per cell area in fibroblasts using immunofluorescence. The p-62 aggregates are negatively correlated with autophagic activity [26]. At baseline, a significant reduction in p-62 aggregates was seen in PD-302, PD-216, and PD-212 fibroblasts compared to the controls (CT: 0.022 ± 0.01 vs. PD-302: 0.006 ± 0.02 (*p* = 0.001); PD-216: 0.012 ± 0.01 (*p* = 0.008); PD-212: 0.013 ± 0.01 (*p* = 0.01)) (Figure 7A,B). After EBSS treatment, control fibroblasts showed the expected reduction in p-62 aggregates, but this treatment did not seem to activate p-62 degradation in any of the patients. Notably, PD-302 fibroblasts exhibited increased p-62 aggregates after treatment, further confirming the failure to properly activate autophagy under starvation conditions. Overall, these results indicate impaired autophagic flux and/or autophagy induction in patients with variants in lysosomal enzyme-related genes. Interestingly, PD-088, a carrier of biallelic *PRKN* variants, was also found to be unresponsive to EBSS treatment, suggesting that alterations in the autophagy-lysosomal pathway are a common pathological mechanism in PD, as established in the literature [44].

## 3. Discussion

This study provides functional evidence of lysosomal dysfunction in EOPD, especially in patients carrying variants in *GLA* and *GLB1,* which are LSD-related genes. Our findings align with multiple studies supporting the involvement of lysosomal genes in PD [5,10,45], and with previous observations linking lysosomal decline to various neurodegenerative diseases, including PD [5,46]. Among them, *GBA1* is the most well-known genetic risk factor for PD. *GBA1* pathogenic effects are believed to result from impaired lysosomal degradation and subsequent buildup of α-synuclein [47,48].

Specifically, we studied *GLA* and *GLB1* as candidate genes that were identified in WES analysis of three patients. In the *GLA* gene encoding GALA, we found the p.Asp313Tyr variant in two unrelated patients. The p.Asp313Tyr is a controversial *GLA* variant that has been reclassified from LP to B, likely due to its high frequency in the population, exceeding the threshold set by ACMG criteria. However, since *GLA* is an X-linked gene, this could explain its high prevalence in healthy control populations. Supporting its potential pathogenicity, p.Asp313Tyr has been reported to be overrepresented in multiple studies in PD patients (Appendix A) [14,20,49,50], and its impact on GALA structure is predicted by in silico studies. The other lysosomal gene identified was *GLB1* encoding β-GAL. The variant *GLB1*:p.Arg419Gln is classified as LP, would also influence the structure of β-GAL. Similarly, variants in *GLB1* have been linked to early-onset dystonia-parkinsonism phenotypes in patient cohorts [21,22].

Functional studies conducted on fibroblasts from patient PD-302 (*GLA*:p.Asp313Tyr) revealed cellular defects associated with lysosomal and autophagic dysfunction. GALA showed partial retention in the Golgi apparatus, likely due to impaired transport to lysosomes, where it performs its hydrolytic function. These lysosomal enzyme transport defects may contribute to the pathophysiology of lysosomal diseases [51]. Additionally, we found abnormalities in lysosomal network morphology, an increased lysosomal pH, and abnormal expression of markers such as p-62 and LC3I/II. Fibroblasts from patient PD-212 (*GBA1*:p.Leu483Pro), although showing morphological abnormalities in the lysosomal network and marker expression, had more acidic lysosomal pH. The defective lysosomal pH in the patients’ fibroblasts align with previous studies, which indicate that the pH range for optimal lysosomal enzyme activity is narrow [52,53]. Additionally, while GALA activity in leukocytes from patient PD-302 remained within the normal range, GCase activity was elevated. This is especially noteworthy since most studies usually report decreased GCase activity in PD patients, regardless of *GBA1* genotype [54]. It suggests that the increased GCase activity observed in our patient may be a compensatory response, reflecting a feedback mechanism between lysosomal enzymes.

From a clinical perspective, it is significant that patient PD-302 presents with demyelinating lesions in the white matter, as previously documented in other patients carrying the *GLA*:p.Asp313Tyr variant. Interestingly, these white matter lesions have been linked to altered GCase activity [55]. Therefore, it is tempting to propose that other patients with the *GLA*:p.Asp313Tyr variant may also exhibit altered GCase activity.

Functional studies of patient PD-216 (*GLB1*:p.Arg419Gln) revealed changes in Golgi morphology, alterations in lysosomal network structure and porosity, an increase in lysosomal pH, and disrupted autophagic flux, suggesting both Golgi and lysosomal dysfunctions. It is important to note that *GLB1*:p.Arg419Gln is an inherited variant, and the *GLB1* gene is linked to a recessive condition. We propose that, while this variant may contribute to the genetic predisposition of patient PD-216, other genetic risk factors—such as the VUS found in *VPS33B*, which is involved in Golgi-to-lysosome trafficking and inherited from the mother—are likely facilitating the disease expression in this patient and causing the Golgi fragmentation observed in the fibroblasts. Additionally, VPS33B plays a role in sorting molecules within cells into various organelles, including lysosomes [56]. As a result, abnormalities in this protein may lead to further damage to lysosomal function.

Overall, this study identified significant changes in lysosomal size, distribution, and pH across all patients with lysosomal-related Parkinson’s disease (PD-302, PD-216, and PD-212). These abnormalities could impact disease mechanisms, as proper lysosomal positioning and acidification are vital for cell health and are tightly regulated by the cell’s internal environment [57]. Indeed, changes in lysosomal morphology and pH affect its function. A pH that is too alkaline or too acidic reduces the activity of hydrolase enzymes [52], while lysosome enlargement—often caused by undigested content—disrupts normal functions, leading to more protein entrapment and faulty trafficking and clearance [58]. Because α-synuclein is primarily degraded in lysosomes [12], any disruption in lysosomal structure or function might lead to its accumulation, impairing autophagy and promoting neurodegeneration. These processes may explain Lewy body formation in EOPD, especially in individuals with lysosomal genetic mutations. Our findings align with existing research on patients with *PRKN* mutations, where lysosomal function remains largely intact and Lewy bodies are sometimes absent [59,60]. These differences support the notion that lysosomal health has a significant influence on PD development and may account for the diverse clinical presentations in PD. In one case, functional studies of patient PD-088 (*PRKN*:p.Trp445*;DEL exon 3-6) showed a preserved lysosomal network despite defects in autophagic flux. Prior research links *PRKN* to early steps in mitophagy [61], and loss of its function can indirectly lead to mitochondrial and autophagy issues.

Lastly, our results also endorse an oligogenic inheritance model for PD where rare inherited variants could act additively or synergistically. Emerging evidence suggests that PD may be influenced by a cumulative genetic burden involving rare variants across known disease-associated genes, rather than by single, highly penetrant mutations. An Italian study reported that approximately 20% of PD cases carried two or more rare exonic variants within a panel of 26 candidate genes [62]. Larger burden analyses using WES have also confirmed enrichment of rare variants in key genes such as *GBA1* and *LRRK2* [63]. In addition, analysis of singleton loss-of-function variants in the *PPMI* cohort revealed a significant exome-wide burden, which improved predictive accuracy when combined with data of common variants [64].

The limitations of this study mainly concern the diagnostic reliability of WES. A key challenge is that it only captures annotated exonic regions and omits non-coding, structural, or regulatory variants such as CNVs, translocations, and inversions, which could also contribute to PD development. Additionally, the high rate of VUS complicates gene prioritization and raises the risk of misclassification, making multidisciplinary review labor-intensive and challenging to scale, especially with large gene panels. Furthermore, functional validation of novel or complex variants often requires multi-omics approaches or experimental models, which are rarely feasible due to limited resources. Overall, these limitations underscore the need for more efficient and scalable strategies for variant interpretation. Regardless of the method employed, functional genomics offers valuable insights into the complex relationships between genotype and phenotype in PD. Combining genetic and cellular data enables researchers to identify new genes and pathways related to PD, deepening our understanding of this diverse set of disorders. Ultimately, these findings are expected to lead to improved diagnosis, prognosis, and treatment options for early-onset Parkinsonian syndromes.

## 4. Materials and Methods

### 4.1. Ethics Statement

All procedures complied with the ethics guidelines and were approved Clinical Research Ethics Committee by Sant Joan de Déu Research Institute and Children’s Hospital (reference PIC-159-22; approval date: 3 February 2023). Written informed consent was obtained from all the participants.

### 4.2. Whole Exome Sequencing

Genomic DNA was extracted from peripheral blood using standard methods. Singleton and trio-Whole exome sequencing (WES) was performed on progenitors and patient’s DNA samples using the DNA Prep with Enrichment (Illumina, San Diego, CA, USA) for library preparation and capture according to the manufacturer’s protocol on the NextSeq500 platform (Illumina, San Diego, CA, USA).

#### 4.2.1. Variant Calling and Annotation

The quality of the reads was determined using FastQC (v.0.11.5). Adapters and low-quality reads were removed with cutadapt (v1.1), and the remaining reads were aligned to the human reference genome GRCh37/hg19 using BWA (v.0.7.15 bwa-mem). Reads with low mapping values and duplicates were removed using BEDtools (v.2.26.0) and Picard (v2.9.0). Variant calling was performed using three callers: GATK (v.3.7), DeepVariant (v.0.10), and Octopus (v.0.6.3-β).

#### 4.2.2. Variant Filtering and Priorization

Variant filtering was performed using Ayakashi.z 3.0, an in-house software developed at the Bioinformatics Unit, Department of Genetic Medicine, Hospital Sant Joan de Déu (HSJD), Barcelona, Spain. Variants were filtered by number of callers (>2), European non-Finnish MAF < 0.01 from gnomAD [65] and pathogenicity predictor CADD score >15 as the main filtering criteria. Intronic variants and synonymous variants with CADD score >15 were evaluated. Subsequently, variants were filtered by a PD gene list. Following the variant filtering, variant prioritization and ranking using knowledge-driven analysis (KDA) for a custom score settling were performed [66] (Appendix A). The aim of this analysis was to assign a total score for each variant based on different qualitative and quantitative criteria. A high score indicates that a variant or gene could be a candidate for further studies. The criteria were the following: variant rarity from gnomAD (v2.1.1); variant classification according to the American College of Medical Genetics guidelines (ACMG) [37] using Franklin website (https://franklin.genoox.com/clinical-db/home); gene tolerance to mutations (z-score to missense variants and pLI score for loss of function variants) from gnomAD (v2.1.1); gene expression in a relevant tissue/organ related to the disease from the Human Protein Atlas (www.proteinatlas.org); a conservation predictor, Genomic Evolutionary Rate Profiling (GERP++) [67]; and pathogenicity predictors: CADD [68], MutationTaster2 [69], Sorting Intolerant From Tolerant (SIFT) [70], Functional Analysis Through Hidden Markov Models (FATHMM-MKL) [71], Polymorphism Phenotyping v2 (PolyPhen2) [72], and SpliceAI [73]. A color scale from deep red (highest value: 15) to green (lowest value: 3) was established for each of the variants analyzed by the KDA. Following the KDA, variants classified by Franklin website as benign (B) or likely benign (LB) with a polymorphic frequency (greater than 1%) in other populations, were excluded from subsequent analyses. Population frequencies of these B/LB variants were verified using the gnomAD (v4.1.0) database.

Candidate variants were validated by Sanger sequencing and visualized using the Integrative Genomics Viewer (IGV).

### 4.3. Skin Biopsy and Fibroblast Culture

Skin biopsies were obtained from the patients using standard clinical procedures. Healthy controls fibroblasts were sourced from the Sant Joan de Déu Children’s Hospital Biobank. Fibroblasts were cultured in DMEM high-glucose (Sigma-Aldrich, St. Louis, MO, USA) supplemented with 10% (*v*/*v*) fetal bovine serum (FBS; Sigma-Aldrich), 2 mM L-glutamine (Sigma-Aldrich) and 100 mg/mL penicillin-streptomycin (Sigma-Aldrich) at 37 °C in a 5% CO_2_ incubator. Routine testing for *Mycoplasma* contamination was performed.

#### Treatments

For the induction of autophagy by amino acid starvation fibroblasts were washed three times with phosphate-buffered saline (PBS) and cultured in Earle’s Balanced Salt Solution (EBSS, Thermo Fisher Scientific, Waltham, MA, USA) for 4 at 37 °C.

### 4.4. Western Blot Assays

Fibroblasts were homogenized in lysis buffer (50 mM Tris HCl pH 7.4, 1.5 mM MgCl_2_, 5 mM EDTA, 1% Triton X-100, 50 mM NaF, and 1 mM Na_2_VO_3_) supplemented with protease inhibitor cocktail (Complete Mini-Protease Inhibitor Cocktail, Roche, Basel, Switzerland). Protein samples (30 µg) were resolved to sodium dodecyl sulfate polyacrylamide gels (SDS-PAGE) and transferred onto PVDF membranes (GE Healthcare, Chicago, IL, USA). The membranes were blocked with 5% defatted-milk in TBST buffer (25 mM Tris, 50 mM NaCl, 2.5 mM KCl, 0.1% Tween-20). Afterwards, the membranes were blotted with specific primary antibodies α-GLA rabbit monoclonal (1:1000; Abcam, Waltham, MA, USA; 168341), α-LAMP1 rabbit polyclonal (1:500; Sig:ma-Aldrich, St. Louis, MO, USA; L1418), α-p-62 rabbit polyclonal (1:500; Sigma-Aldrich; P0067), α-LC3 rabbit polyclonal contributo (1:500; Sigma-Aldrich; L7543), and α-β-Actin mouse monoclonal (1:8000; Sigma-Aldrich; A5316) which were detected using secondary antibodies coupled to horseradish peroxidase. A standard curve rhGLA protein (R&D systems, Minneapolis, MN, USA; 6146-GH) was performed in the α-GLA membrane. The chemiluminescence signal was visualized by iBright™ CL1000 Imaging System (Thermo Fisher). Bands intensity was measured using ImageJ/Fiji software v.1.52p (National Institutes of Health, Bethesda, MD, USA).

### 4.5. Immunofluorescence

Fibroblasts were seeded onto glass coverslips, washed with PBS and fixed in pre-warmed 4% paraformaldehyde (PFA) for 20 min at room temperature. Cells were permeabilized with 0.2% Triton X-100 in PBS for 30 min at room temperature and were blocked with 1% bovine serum albumin (BSA) and 4% serum in PBS for 1 h at room temperature. The specific primary antibodies α-GLA rabbit polyclonal (1:100; Biorbyt, Cambridge, UK; orb167113), α-TGN46 sheep antibody (1:100; Bio-Rad, Hercules, CA, USA; AHP500GT), α-LAMP1 mouse antibody (1:100; Developmental Studies Hybridoma Bank, University of Iowa, Iowa City, IA, USA; H4A3), and α-p-62 rabbit polyclonal (1:200; Sigma-Aldrich, St. Louis, MO, USA; P0067) were incubated overnight at 4 °C. The secondary conjugated antibodies Alexa Fluor^®^ 488 or Alexa Fluor^®^ 633-labeled (1:500; Thermo Fisher Scientific, Waltham, MA, USA) were incubated for 2 h at room temperature. The coverslips were mounted with Fluoromont-G with DAPI (4′,6-diamidino-2-phenylindole) (Thermo Fisher Scientific, Waltham, MA, USA).

### 4.6. Lysosomal pH Analysis

To assess lysosomal alkalinization, cells were seeded onto µ-Dish 35 mm dishes (Ibidi), washed with PBS, and incubated with 1 μM LysoSensor™ Green DND-189 (Thermo Fisher Scientific, Waltham, MA, USA, L7535) for 30 min at 37 °C.

### 4.7. Image Acquisition and Super-Resolution Images Analysis

Images were acquired with a Leica TCS SP8 X White Light Laser confocal microscope with Hybrid spectral detectors and HyVolution (Leica Microsystems, Wetzlar, Germany) using the Leica LAS X software (version 3.1.5). Z-stacks consisting of 15 sections were acquired at intervals of 0.25 μm throughout the sample’s thickness. Appropriate negative controls were used to adjust confocal settings to avoid non-specific fluorescence artifacts. The images were captured with an 8-bit depth per pixel, and the pinhole diameter was set at 0.7 AU. Subsequent image deconvolution was performed using the Huygens Professional software v.17.10.0p7 64b (SVI). Identical confocal settings were applied during image acquisition for comparative analysis. Maximum intensity projections were generated utilizing LAS X software v.5.1.0.25593 (Leica Microsystems, Wetzlar, Germany).

#### 4.7.1. Colocalization Analysis

Colocalization was quantified using Manders’ Overall Overlap Coefficient (MOC), which measures the proportion of the total fluorescence signal in one channel that overlaps with the signal in a second channel. Unlike simple area-based overlap metrics, MOC is less affected by background noise, as it gives greater weight to pixels with higher intensity values, providing a more reliable assessment of true signal colocalization.

#### 4.7.2. Lysosomes Analysis

The diameter of the lysosomes was calculated as the distance between peaks in the intensity profile. Furthermore, a porosity study was performed to determine the percentage of area occupied by pores within multiple regions of interest; each sized 50 × 50 pixels, generated across the entire lysosomal network within the cell.

#### 4.7.3. Golgi Apparatus Analysis

Golgi fragmentation was assessed through several quantitative measures. The distance between the nucleus and the farthest point of the Golgi was analyzed [74]. This distance was calculated by establishing a line connecting the farthest point of the Golgi with the centroid of the nucleus, followed by measuring the distance between this extremity of the Golgi and the nearest point on the nucleus. Moreover, the number of Golgi fragments per cell was determined [75]. These analyses were executed using Python (v.3.7.4) and ImageJ/Fiji software v.1.52p (NIH).

### 4.8. Statistical Analysis

Data were presented as mean ± standard deviation (SD) or as box plots representing the median, the 25th and 75th percentiles (box edges), and the minimum and maximum values (whiskers). Data normality was assessed using the Shapiro–Wilk test. The specific statistical tests used are detailed in each figure legend. *p*-values less than 0.05 were considered statistically significant, with the significance levels indicated as follows: * *p* < 0.05, ** *p* < 0.01 and, *** *p* < 0.001. All statistical analyses and graphical representation were performed using the GraphPad Prism v.8.0.1 (GraphPad Software Inc., San Diego, CA, USA).

## 5. Conclusions

Our results support the idea that lysosomal dysfunction is key in the development of EOPD. On a molecular level, changes seen in lysosomal shape, luminal pH, and autophagic flux indicate a broad disruption of the lysosomal-autophagic system in patients with variants in lysosomal hydrolase genes like *GLA*, *GLB1*, and *GBA1*. Genetically, the WES data hint at a possible oligogenic inheritance in some undiagnosed EOPD patients, where both inherited and new variants can combine, with each patient potentially having a unique variant profile that influences their disease presentation. These findings emphasize the need to include functional genomics in understanding the genetic and biological mechanisms of EOPD. Additionally, studying rare lysosomal disorders linked to *GLA, GLB1* and *GBA1* variants can shed light on PD, the most common movement disorder. This could reveal shared pathways involving impaired lysosomal function and disrupted protein clearance, which may help uncover potential uses for lysosomal pathways in treating EOPD and PD.

## Figures and Tables

**Figure 1 ijms-26-09454-f001:**
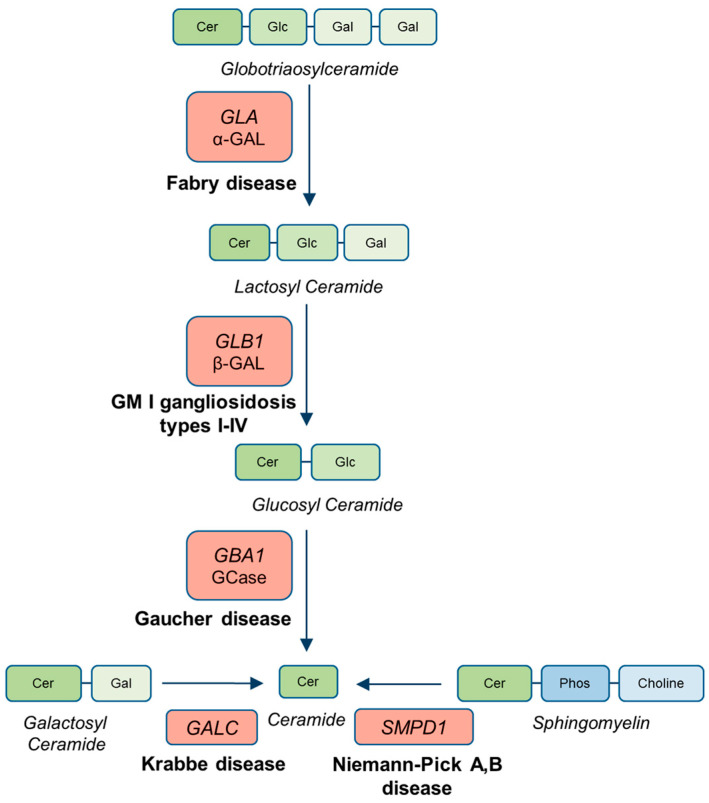
Sphingolipid metabolism pathway. Hydrolytic enzymes involved in lysosomal sphingolipid metabolism are highlighted by their corresponding gene names (red boxes). The ceramides and their metabolites (green boxes) and choline and its metabolite (blue boxes) are also depicted. For the genes analyzed in this study, the enzyme names are shown beneath the gene symbols.

**Figure 2 ijms-26-09454-f002:**
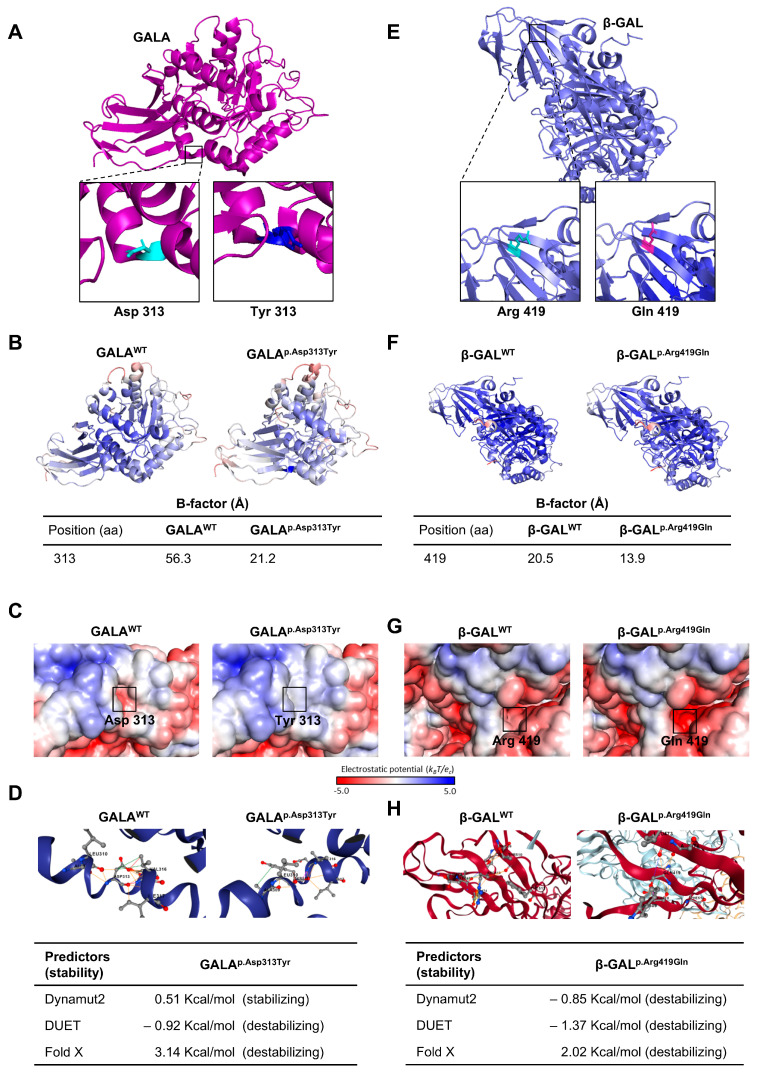
In silico structural analysis of the GALA^p.Asp313Tyr^ and β-GAL^p.Arg419Gln^. (**A**) Structural representation of GALA based on PDB:1R46. Detailed views show the native Asp313 (left panel) and the mutant Tyr313 (right panel), obtained by Mutagenesis Wizard tool in PyMOL v.2.5.7 (Schrödinger, LLC, New York, NY, USA; https://www.pymol.org/) (**B**) B-factor analysis comparing GALA^WT^ and the GALA^p.Asp313Tyr^, performed in PyMOL. (**C**) Electrostatic surfaces potential of GALA^WT^ and the GALA^p.Asp313Tyr^ calculated with the APBS module in PyMOL. A red-white-blue scale ranging from -5/+5 is used, with red and blue colors indicating negative and positive electrostatic potentials, respectively. (**D**) Predicted effects on protein stability for GALA^p.Asp313Tyr^ using DynaMut2, DUET, and FoldX. (**E**) Structural representation of the β-GAL based on PDB:3THC, showing native Arg419 (left panel) and mutant Gln419 (right panel). (**F**) B-factor analysis of β-GAL^WT^ and β-GAL^p.Arg419Gln^ variant. (**G**) Electrostatic surface potential for wild-type β-GAL^WT^ and β-GAL^p.Arg419Gln^ calculated by APBS in PyMOL. (**H**) Predicted scores from protein stability predictors for the β-GAL^p.Arg419Gln^ (DynaMut2, DUET and FoldX).

**Figure 3 ijms-26-09454-f003:**
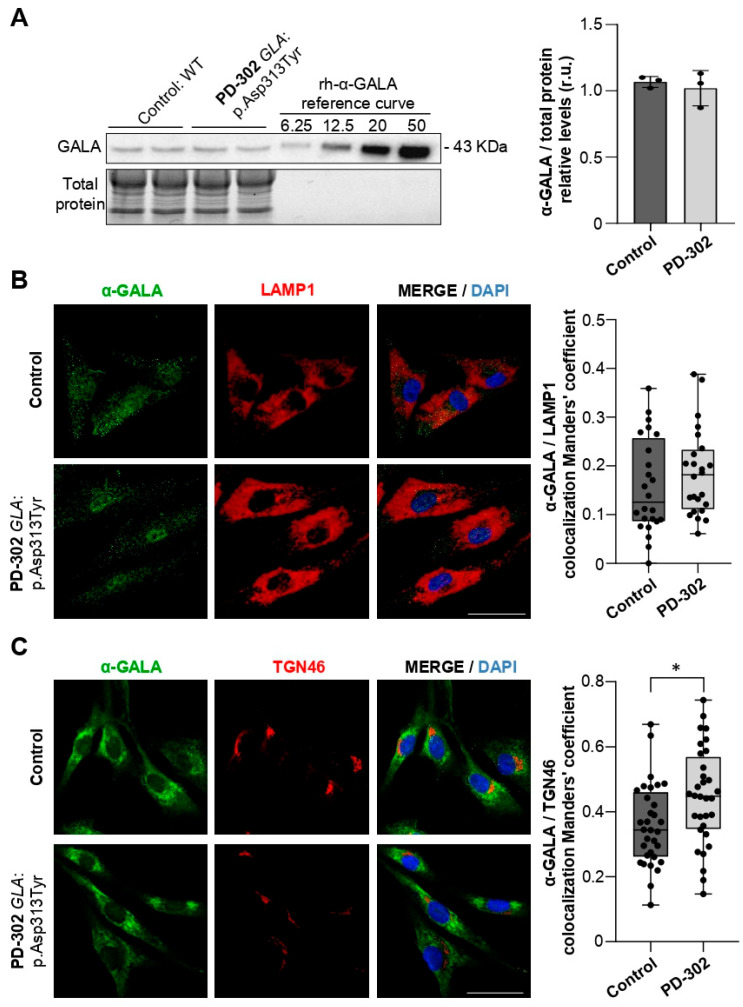
GALA abnormal retention in the trans-Golgi network in the PD-302 fibroblasts. (**A**) Western blot analyses of GALA in PD-302 and control fibroblasts. Cell lysates from control (lanes 1–2) and patients’ (lanes 3–4) samples were loaded in duplicates. Known quantities (6.25, 12.5, 20 and 50 ng) of recombinant-human-GALA were loaded as internal control in lanes 5-8. Lower panel of the image represent the electrophoresis gel corresponding to the Western blot in the upper panel. The quantification of unknown samples (ng of GALA/total protein) was obtained by dividing the absolute amount of GALA (ng) in each band, as extrapolated for from the standard curve for the amount of total protein in the samples, assessed by BCA assay. (mean ± SD; *n* = 3 independent biological replicates). Student *t*-test. (**B**) Representative immunofluorescence images of GALA (green), LAMP1 (red) and nuclei (DAPI, blue) in PD-302 and control fibroblasts. Colocalization analysis of GALA/LAMP1 using Manders’ coefficient (box plot (25th percentile, median, 75th percentile; whiskers represent minimum and maximum values); *n* = 3 independent biological replicates). Mann–Whitney *U*-test. (**C**) Representative immunofluorescence images of GALA (green), TGN46 (red) and nuclei (DAPI, blue) in PD-302 and control fibroblasts. Quantitative colocalization analysis of GALA/TGN46 using Manders’ coefficient (box plot as in B; *n* = 3 independent biological replicates). Mann–Whitney *U*-test (* *p* < 0.05). Scale bars: 50 µm. Abbreviation: r.u., relative units.

**Figure 4 ijms-26-09454-f004:**
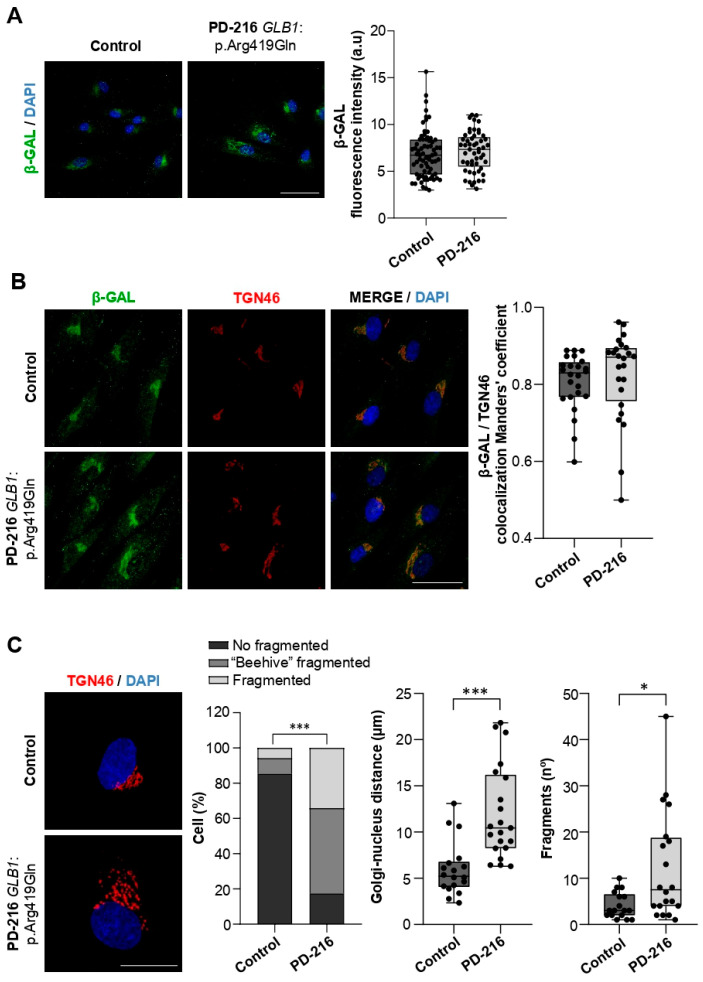
β-GAL levels are preserved, but Golgi apparatus morphology is altered in PD-216 fibroblasts. (**A**) Representative immunofluorescence images of β-galactosidase (β-GAL, green) and nuclei (DAPI, blue) in patient (PD-216, *GLB1*:p.Arg419Gln) and control fibroblasts. Quantification of **β-GAL** fluorescence intensity is shown as a box plot (25th percentile, median, 75th percentile; whiskers represent minimum and maximum values; *n* = 3 independent biological replicates). Mann–Whitney *U*-test. (**B**) Representative immunofluorescence images of β-GAL (green), TGN46 (red) and nuclei (DAPI, blue) in PD-216 and control fibroblasts. Quantitative colocalization analysis of β-GAL/TGN46 using Manders’ coefficient (box plot as in A; *n* = 3 independent biological replicates). Mann–Whitney *U*-test. (**C**) Representative immunofluorescence images TGN46 (red) and nuclei (DAPI, blue) in PD-216 and control fibroblasts. Golgi morphology was assessed by classification of structural phenotypes (percentage shown in bar plot; *n* = 3 independent biological replicates; ≥100 randomly selected cells per replicate). χ^2^ (*** *p* < 0.001). Quantitative analyses of Golgi-nucleus distance and Golgi fragmentation are shown as box plots (as in A; *n* = 3 independent biological replicates). Mann–Whitney *U*-test (* *p* < 0.05; *** *p* < 0.001). Scale bars: 50 µm. Abbreviation: a.u, arbitrary units.

**Figure 5 ijms-26-09454-f005:**
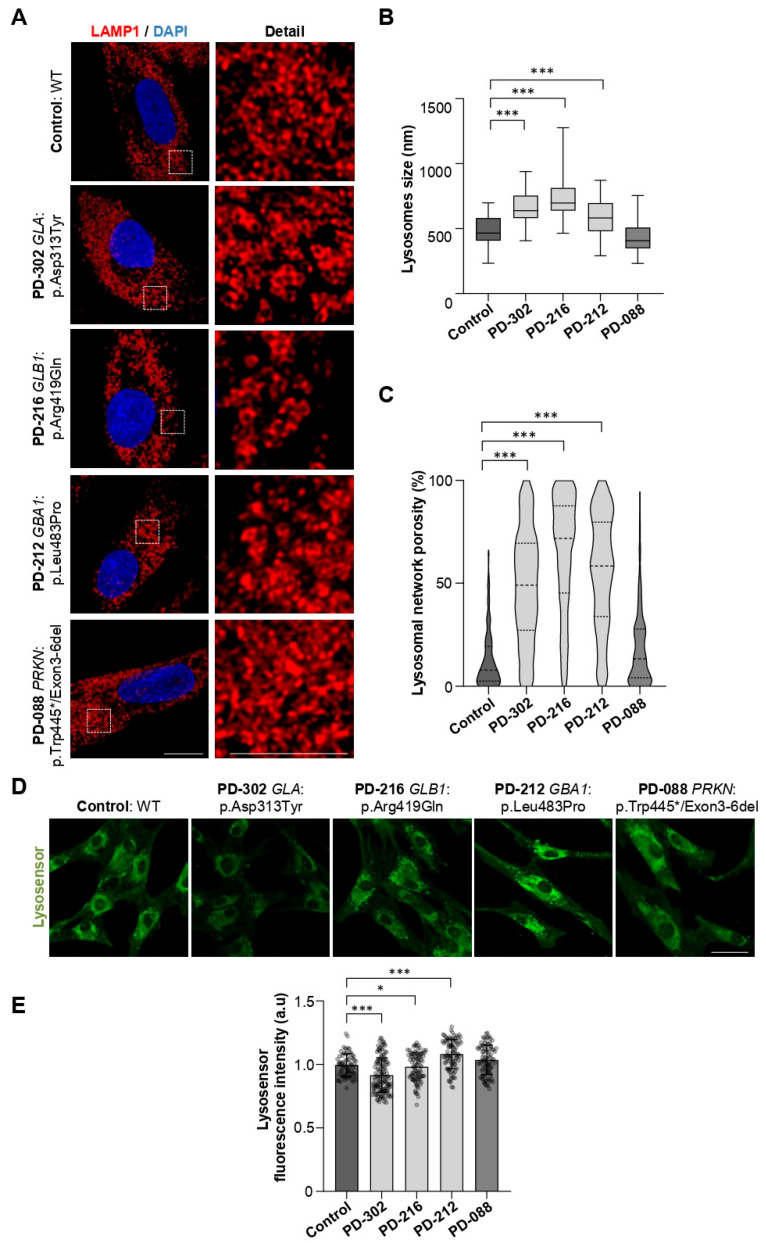
Patient-derived fibroblasts with genetic variants in lysosomal genes presented an altered pH and lysosomal morphology compared to PD patients with variants in other non-lysosomal genes and to the control. (**A**) Representative super-resolution images of LAMP1 (red) and nuclei (DAPI, blue) in patients with lysosomal affectation (PD-302, *GLA*:p.Asp313Tyr; PD-216, *GLB1*:p.Arg419Gln; and PD-212, *GBA1*:p.Leu483Pro), patient without variants in lysosomal genes (PD-088, *PRKN*:p.Trp445*;DEL exon 3-6) and control fibroblasts. (**B**) Quantification of lysosome size (nm) is shown as a box plot (25th percentile, median, 75th percentile; whiskers represent minimum and maximum values; *n* = 3 independent biological replicates; ≥80 randomly selected lysosomes per replicate). Kruskal–Wallis followed by the Dunn’s test for multiple comparisons were applied (* *p* < 0.05; *** *p* < 0.001). (**C**) Quantification of the percentage of porosity in the lysosomal network is shown in violin plots (thick dashed line represents the median, and the softer dashed lines represent the 25th and 75th percentiles; *n* = 3 independent biological replicates). Kruskal–Wallis followed by the Dunn’s test for multiple comparisons (*** *p* < 0.001). (**D**) In vivo representative images of pH-dependent lysosome staining LysoSensor green in patients and control fibroblasts. (**E**) Quantification of Lysosensor Green fluorescence intensity (mean ± SD; *n* = 5 independent biological replicates). One-way ANOVA followed by Tukey’s test for multiple comparisons (* *p* < 0.05; *** *p* < 0.001). Scale bars: 50 µm. Abbreviation: a.u., arbitrary units.

**Figure 6 ijms-26-09454-f006:**
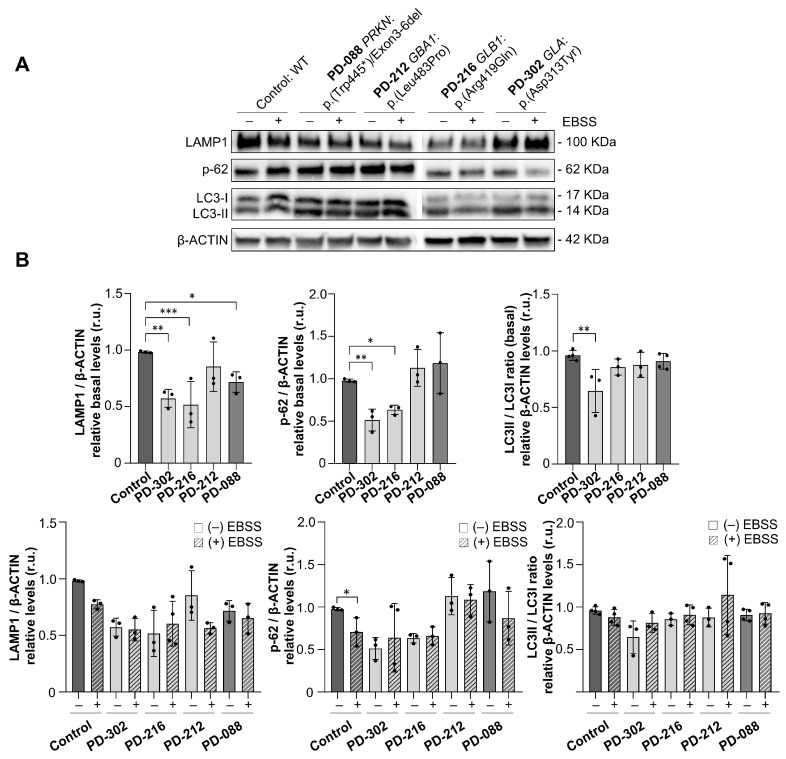
Impaired autophagy is exacerbated in patients carrying variants in lysosomal genes. (**A**) Western blot analysis of LAMP1, p-62, LC3-I, and LC3-II proteins in fibroblasts from patients with lysosomal gene variants (PD-302, *GLA*:p.Asp313Tyr; PD-216, *GLB1*:p.Arg419Gln; and PD-212, *GBA1*:p.(Leu483Pro)), a patient without lysosomal genes variants (PD-088, *PRKN*:p.Trp445*;DEL exon 3-6), and healthy control fibroblasts. Samples were analyzed under basal conditions (–) and following EBSS-induced autophagy (+). β-ACTIN was used as a loading control. (**B**) Quantification of protein levels under basal conditions (upper panel) and comparison of protein levels between basal and EBSS-treated conditions (lower panel) (mean ± SD; from at least 3 independent biological replicates). One-way ANOVA followed by Tukey’s multiple comparisons test (* *p* < 0.05; ** *p* < 0.01; *** *p* < 0.001). Abbreviation: r.u., relative units.

**Figure 7 ijms-26-09454-f007:**
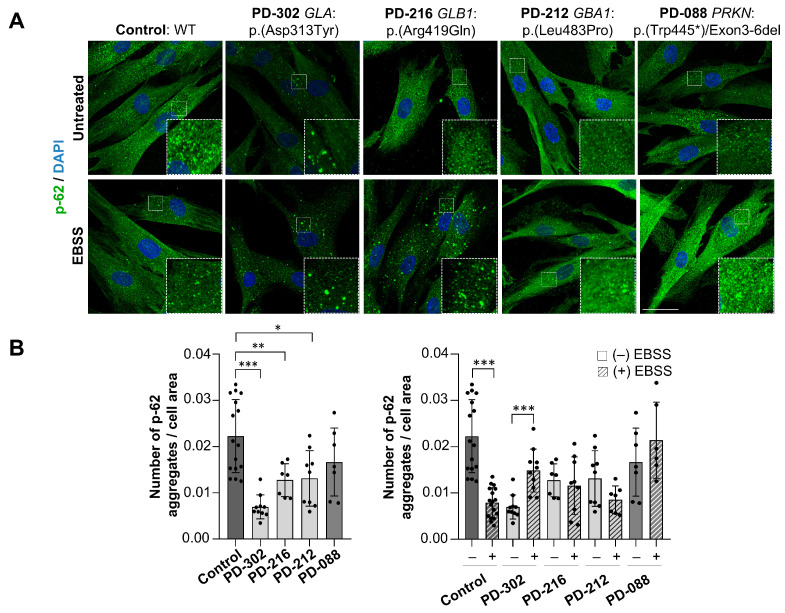
Aberrant response to autophagy induction in patients’ fibroblasts with lysosomal genes variant. (**A**) Representative immunofluorescence images of p-62 in untreated and EBSS-treated fibroblasts. A magnified detail is shown in each image. Scale bars: 50 μm. (**B**) Quantification of p-62 aggregates per cell area in untreated and EBSS-treated fibroblasts (mean ± SD; *n* = 3 independent biological replicates and individual values are displayed as dots). One-way ANOVA followed by Tukey’s multiple comparisons test (* *p* < 0.05; ** *p* < 0.01; *** *p* < 0.001).

**Table 1 ijms-26-09454-t001:** Detailed information of genetic variants prioritized in patients PD-302, PD-227, and PD-216 through WES.

P	General Information							Scores											
	Gene/ MIM Phenotype	Pathways	Variant	INH	ACMG	CADD	Freq.	ACMG	CADD	Freq.	MT	PolP2	SIFT	F-MKL	S-AI	GERP	Z/pLI	BE	T
**PD-302 (WES-Trio)**	*FBXO7/* *PARK 15 (AR)*	LY/ATP	p.(Ala248Thr) c.742G>A	M	VUS	24.5	3.97 × 10^−6^	2	1	3	1	1	1	1	0	1	0	1	12
	*LRRK2/* *PARK 8 (AD)*	LY/VT/NR	p.(Phe1436Leu) c.4306T>C	M	VUS	24.1	NF	2	1	3	1	1	0	1	0	1	0	1	11
	***GLA/*** **FD *(XL)*** ^&^	**LY/ATP**	**p.Asp313Tyr ** **c.937G>T**	**M**	**B**	**18.3**	**4.46 × 10^−3^**	**1**	**1**	**1**	**1**	**1**	**1**	**1**	**0**	**0**	**0**	**1**	**8**
**PD-227 (WES-Singleton)**	*ABL2/* *-*	NR	p.(Arg180Cys) c.538C>T	.	LP	34.0	8.79 × 10^−6^	3	2	2	1	1	1	1	0	0	0	1	12
	***GLA/*** **FD *(XL)*** ^&^	**LY/ATP**	**p.Asp313Tyr ** **c.937G>T**	**M**	**B**	**18.3**	**4.46 × 10^−3^**	**1**	**1**	**1**	**1**	**1**	**1**	**1**	**0**	**0**	**0**	**1**	**8**
	*BORCS8/* *NDOABA (AR)*	LY/MAPK	p.(Arg85His) c.254G>A	.	VUS	22.8	1.35 × 10^−5^	2	1	2	1	0	0	0	0	1	0	1	8
**PD-216 (WES-Trio)**	*VPS33B/(AR) **	LY/GLG/ ATP	p.(Arg398Cys) c.1192C>T	M	VUS	33.0	NF	2	2	3	1	1	1	1	0	1	0	1	13
	***GLB1/*** ***(AR)*** ^%^	**LY/ATP/** **NR**	**p.Arg419Gln ** **c.1256G>A**	**F**	**LP**	**35.0**	**7.06 × 10^−6^**	**3**	**2**	**2**	**1**	**0**	**0**	**1**	**1**	**1**	**0**	**1**	**12**
	*DAPK1/* *-*	LY/ATP	p.(Arg666Gln) c.1997G>A	M	VUS	25.4	6.22 × 10^−5^	2	1	2	1	0	1	1	0	1	0	1	10

* MIM Phenotypes for *VPS33B*: Arthrogryposis, renal dysfunction, and cholestasis 1, Cholestasis, progressive familial intrahepatic, Keratoderma-ichthyosis-deafness syndrome; ^%^ MIM Phenotypes for *GLB1*: GM1-gangliosidosis, type I; GM1-gangliosidosis, type II; GM1-gangliosidosis, type III; Mucopolysaccharidosis type IVB (Morquio); ^&^ MIM Phenotype for *GLA*: Fabry disease (FD); P: patient; LY: Lysosome; ATP: Autophagy; GLG: Golgi apparatus; MIT: Mitochondria; ER: Endoplasmic reticulum; MAPK: Mitogen-Activated Protein Kinase; VT: Vesicular Trafficking; NR: Neuronal; INH: Inheritance; M: Mother; F: Father; ACMG: American College of Medical Genetics and Genomics; VUS: Variant of Uncertain Significance; B: Benign; LP: Likely Pathogenic; NF: Not Found in gnomAD; CADD: Combined Annotation Dependent Depletion; Freq: Allele frequency; MT: MutationTaster; PolP2: PolyPhen-2; NDOABA: infantile-onset neurodegeneration with optic atrophy and brain abnormalities; SIFT: Sorting Intolerant From Tolerant; F-MKL: FATHM-MKL; GERP: Genomic Evolutionary Rate Profiling; S-AI: SpliceAI; Z/pLI: Z-Loss-of-function; BE: Brain expression level; T: Total score.

## Data Availability

All relevant data that support the findings of this study are presented in the main text and Appendix A. Other source data are available from the corresponding authors upon reasonable request.

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
