# Peer review of "Lysosomal Network Defects in Early-Onset Parkinson’s Disease Patients Carrying Rare Variants in Lysosomal Hydrolytic Enzyme Genes"

_ijms, 2025, doi:10.3390/ijms26199454_

Round 1

Reviewer 1 Report

Comments and Suggestions for Authors

1. In the introduction section of the article, it is emphasized that "Lysosomes are essential for maintaining neuronal proteostasis by degrading cellular debris through autophagy and endocytosis." The changes in autophagic lysosomal activity refer to the internal changes within nerve cells, and whether the results obtained from exome sequencing using patient blood are directly related to the early Parkinson's disease caused by damage to the autophagic lysosomal system? What is the significance? It is suggested to provide relevant evidence.
2. Based on the results of exome sequencing, further experimental verification should be conducted. Why were fibroblasts used instead of nerve cells for more intuitive observation of the changes in autophagic lysosomal activity and the resulting nerve cell damage? These existing results are difficult to support the main theme of the article. It is suggested to provide relevant evidence.
3. The conclusion of the article claims that "Patients carrying variations in lysosomal hydrolase genes such as GLA, GLB1, and GBA1 have extensive damage to the lysosomal-autophagic system, and this functional disorder may lead to the accumulation of α-synuclein and promote the neurodegenerative process of Parkinson's disease." However, no direct experimental evidence is provided to support this claim. It is suggested to supplement.
4. Although the article points out that lysosomal abnormalities may affect the degradation of α-synuclein, it does not further explain "how the size and pH changes of lysosomes specifically lead to protein accumulation" by combining existing research (such as the mechanism of the association between lysosomal function and α-synuclein metabolism cited in the references). It is suggested to deepen the discussion on the mechanism and combine literature evidence in the discussion section.
5. The reference format should be unified and mainly include those from the past five years.

Author Response

Reviewer 1

  1. In the introduction section of the article, it is emphasized that "Lysosomes are essential for maintaining neuronal proteostasis by degrading cellular debris through autophagy and endocytosis." The changes in autophagic lysosomal activity refer to the internal changes within nerve cells, and whether the results obtained from exome sequencing using patient blood are directly related to the early Parkinson's disease caused by damage to the autophagic lysosomal system? What is the significance? It is suggested to provide relevant evidence.

The exome results come from three patients out of 49 who had variants in genes that encode lysosomal hydrolases. We aimed to provide functional evidence of lysosomal dysfunction in EOPD, particularly in patients carrying variants in GLA and GLB1, which encode lysosomal hydrolases associated with lysosomal storage disorders. GBA1 is a major risk factor for PD, and numerous reports have utilized patient-derived cellular models to demonstrate that the pathogenic GBA1 variant affects lysosomal function in PD patients.

Our study used functional genomics to investigate how variants in other lysosomal hydrolase genes affect cellular phenotypes in fibroblasts from patients with EOPD. Therefore, defining other lysosomal genes may contribute to improved diagnostic outcomes in some patients. For example, regarding the GLA variant present in some PD patient cohorts, this study will provide further pathophysiological insight into its potential role in the development of PD. Furthermore, from a therapeutic perspective, studying lysosomes in patients carrying variants in lysosomal genes could shed light on the potential use of lysosomal pathways for treating EOPD and PD.

We have added the following final sentence in the abstract: “which could enhance both diagnosis and future treatments”. We also modified the conclusion accordingly (lines 691-693).

  1. Based on the results of exome sequencing, further experimental verification should be conducted. Why were fibroblasts used instead of nerve cells for more intuitive observation of the changes in autophagic lysosomal activity and the resulting nerve cell damage? These existing results are difficult to support the main theme of the article. It is suggested to provide relevant evidence.

We thank the reviewer for this comment. In our experience, the patient-derived fibroblasts have proven extremely useful in the study of many neurological diseases, enabling the detection of the impact of genetic variants on the proteins encoded by genes expressed in this cell type (Pijuan et al Front Neurosci 2022, https://doi.org/10.3389/fnins.2022.784880). Very recently, Moreira-Leite et al  bioRxiv 2025 https://doi.org/10.1101/2025.06.24.661057) reported: “Our findings demonstrate that human fibroblasts faithfully recapitulate lysosomal and mitochondrial dysfunctions characteristic of neurodegenerative diseases. Moreover, the use of robust assays positions these cells as a valuable platform for high-throughput screening to identify novel therapeutics targeting lysosomal and mitochondrial pathways”.

  1. The conclusion of the article claims that "Patients carrying variations in lysosomal hydrolase genes such as GLA, GLB1, and GBA1 have extensive damage to the lysosomal-autophagic system, and this functional disorder may lead to the accumulation of α-synuclein and promote the neurodegenerative process of Parkinson's disease." However, no direct experimental evidence is provided to support this claim. It is suggested to supplement.

We fully agree with the reviewer, and we have modified the conclusion accordingly.

  1. Although the article points out that lysosomal abnormalities may affect the degradation of α-synuclein, it does not further explain "how the size and pH changes of lysosomes specifically lead to protein accumulation" by combining existing research (such as the mechanism of the association between lysosomal function and α-synuclein metabolism cited in the references). It is suggested to deepen the discussion on the mechanism and combine literature evidence in the discussion section.

We thank the reviewer for this comment. We have included the following paragraph in the section discussion (line 517):

“Indeed, changes in lysosomal morphology and pH affect its function. A pH that is too alkaline or too acidic reduces the activity of hydrolase enzymes (Brady RO et al, J Biol Chem, 1965, PMID: 14253443), while lysosome enlargement—often caused by undigested content—disrupts normal functions, leading to more protein entrapment and faulty trafficking and clearance (Scerra G et al, iScience 2021, https://doi.org/10.1016/j.isci.2021.102707)”.

  1. The reference format should be unified and mainly include those from the past five years.

We thank the reviewer for this comment. We have modified some references in the bibliography.

Reviewer 2 Report

Comments and Suggestions for Authors

Pascual and colleagues identified two SNPs that cause early-onset Parkinson's disease in three patients. The authors selected candidate SNPs systematically. Specifically, they focused on lysosomal hydrolytic enzyme genes and studied the influence of variants on the enzyme activity through in silico analysis. In addition, the authors examined fibroblasts derived from the patients. They found the low rates of enzyme exit from the Golgi apparatus toward lysosomes or fragmentation of the Golgi apparatus. They have also reported alteration in lysosomal pH and morphology, suggestive of lysosomal dysfunction. These changes were accompanied by impaired autophagic flux. One of the two variants in this report is the first to be identified as pathogenic. Overall, the study results are informative. However, a couple of points should be clarified.

Major points
All patients inherited the responsible variant from their mother or father, indicating that the mode of inheritance is likely autosomal dominant. The question is why their mother or father did not present the symptoms. It is unlikely that bearing the variants is sufficient for the onset of the disease.

All patients are carrying several other variants with a high-risk score. How can the authors attribute the disease risk only to the lysosomal enzymes? Other variants or a specific combination of variants, including those of lysosomal enzymes, can be more attributable. It is plausible that their mother and father, who are carrying only variants of lysosomal enzymes but not other risk variants, are free from the disease. 

PD-216 inherits a VPS33 variant from his mother and a GLB1 variant from his father. Since VPS33 is involved in Golgi-to-lysosome trafficking, the Golgi fragmentation observed in the fibroblast derived from PD-216 can be attributable not only to a GLB1 variant but also to a VPS33 variant.  This point should be discussed.

Minor points
The presented data indicate that not only an increase in lysosomal pH but also a more acidic condition in lysosomes can relate to the onset of Parkinson's disease. Does the fact suggest that the range of optimal lysosomal pH is narrow and that pH should be strictly controlled? But such a notion does not align with the description "decreased acidity of lysosomes demonstrate a common pattern of lysosomal dysfunction in patients with PD who carry these variants." in lines 373-375.

The second Western blots in "original images" seem not to be related to Figure 7A but to Figure 6A

There is a typo in line 170; dyinvolvsfunction.

There is no explanation of what the right side of the western blot means in Figure 3A. 

Author Response

Reviewer 2

Pascual and colleagues identified two SNPs that cause early-onset Parkinson's disease in three patients. The authors selected candidate SNPs systematically. Specifically, they focused on lysosomal hydrolytic enzyme genes and studied the influence of variants on the enzyme activity through in silico analysis. In addition, the authors examined fibroblasts derived from the patients. They found the low rates of enzyme exit from the Golgi apparatus toward lysosomes or fragmentation of the Golgi apparatus. They have also reported alteration in lysosomal pH and morphology, suggestive of lysosomal dysfunction. These changes were accompanied by impaired autophagic flux. One of the two variants in this report is the first to be identified as pathogenic. Overall, the study results are informative. However, a couple of points should be clarified.

Major points

All patients inherited the responsible variant from their mother or father, indicating that the mode of inheritance is likely autosomal dominant. The question is why their mother or father did not present the symptoms. It is unlikely that bearing the variants is sufficient for the onset of the disease.

All patients are carrying several other variants with a high-risk score. How can the authors attribute the disease risk only to the lysosomal enzymes? Other variants or a specific combination of variants, including those of lysosomal enzymes, can be more attributable. It is plausible that their mother and father, who are carrying only variants of lysosomal enzymes but not other risk variants, are free from the disease. 

We fully agree with the reviewer's perspective and suggest that variants alone are not enough for disease expression. We mentioned this in the abstract (Line 44) “Additional candidate variants were found related to lysosomes, Golgi apparatus, and neurodegeneration, suggesting a multifactorial contribution to the disease.” In the discussion section, we commented (lines 526 to 531) “Lastly, our results also endorse an oligogenic inheritance model for PD where rare inherited variants could act additively or synergistically.”

Regarding the parents of the patients, one of the male patients inherited the p.Asp313Tyr variant in the X-linked GLA gene from his healthy mother. GLA pathogenic variants have been initially related with Fabry disease in males, suggesting recessive X-linked inheritance. More recently, Fabry disease has also been observed in females, which indicates that Fabry disease may segregate as a partially dominant X-linked trait. Thus, the effect of GLA variants can be observed in hemizygous males and in some but not all heterozygous females.  

For the male patient who inherited the p.Arg419Gln variant in the GLB1 gene from his father, which is associated with an autosomal recessive condition, we proposed that carrying a single pathogenic allele increases the risk of PD. As a result, his father would not be affected, but the patient would be, since he also carries other risk variants.

  1. PD-216 inherits a VPS33 variant from his mother and a GLB1 variant from his father. Since VPS33 is involved in Golgi-to-lysosome trafficking, the Golgi fragmentation observed in the fibroblast derived from PD-216 can be attributable not only to a GLB1 variant but also to a VPS33 variant.  This point should be discussed.

We thank the reviewer for this comment. In the discussion section we have included the following (line 508): “We propose that, while this variant may contribute to the genetic predisposition of patient PD-216, other genetic risk factors—such as the VUS found in VPS33B, which is involved in Golgi-to-lysosome trafficking and inherited from the mother—are likely facilitating the disease expression in this patient and causing the Golgi fragmentation observed in the fibroblasts”

Lines 509-512: “Additionally, VPS33B plays a role in sorting molecules within cells into various organelles, including lysosomes (Galmes R et al, Traffic, 2015, https://doi.org/10.1111/tra.12334). As a result, abnormalities in this protein may lead to further damage to lysosomal function.

Minor points

  1. The presented data indicate that not only an increase in lysosomal pH but also a more acidic condition in lysosomes can relate to the onset of Parkinson's disease. Does the fact suggest that the range of optimal lysosomal pH is narrow and that pH should be strictly controlled? But such a notion does not align with the description "decreased acidity of lysosomes demonstrate a common pattern of lysosomal dysfunction in patients with PD who carry these variants." in lines 373-375.

We thank the reviewer for this helpful comment. We have corrected the sentence: “The larger size, higher porosity, and altered acidity of lysosomes demonstrate a common pattern of lysosomal dysfunction in patients with PD who carry these variants.”

Additionally, we have corrected the discussion section (lines 484 to 488) as follows: “Fibroblasts from patient PD-212 (GBA1:p.Leu483Pro), although showing morphological abnormalities in the lysosomal network and marker expression, had more acidic lysosomal pH. The lysosomal pH results of these patients align with previous studies, which indicate that the pH range for optimal lysosomal enzyme activity is narrow (REF).

We have include the following references:

  1. R O Brady, J Kanfer, D Shapiro, J. The metabolism of glucocerebrosides. J Biol Chem. 1965 Jan:240:39-43.PMID: 14253443
  2. Ponsford AH, (2021) Live imaging of intra-lysosome pH in cell lines and primary neuronal culture using a novel genetically encoded biosensor. Autophagy 17:1500–1518. doi:1042 10.1080/15548627.2020.1771858)
  3. The second Western blots in "original images" seem not to be related to Figure 7A but to Figure 6A

We thank the reviewer for this comment. We have corrected the labelling in the original image file.

  1. There is a typo in line 170; dyinvolvsfunction.

We thank the reviewer for this comment. We have corrected dyinvolvsfunction by dysfunction.

  1. There is no explanation of what the right side of the western blot means in Figure 3A. 

We appreciate the reviewer for the opportunity to clarify this point. The first four lanes show control and patient samples, loaded in duplicates. On the right side of the gel, we loaded known amounts of recombinant human α-Galactosidase A protein (6.25, 12.5, 20, and 50 ng) to identify the correct band and perform an absolute quantification of unknown band intensities relative to this standard curve. The protein amount of unknown samples, extrapolated using this standard curve, was further normalized to the total protein loaded, as measured by the BCA assay. The figure legend was modified accordingly.